# Centennial response of Greenland's three largest outlet glaciers

Shfaqat A. Khan [1✉], Anders A. Bjørk[2], Jonathan L. Bamber [3], Mathieu Morlighem [4], Michael Bevis[5], Kurt H. Kjær[6], Jérémie Mouginot [7], Anja Løkkegaard[1], David M. Holland [8], Andy Aschwanden [9], Bao Zhang[10], Veit Helm[11], Niels J. Korsgaard [12], William Colgan[12], Nicolaj K. Larsen[6], Lin Liu [13], Karina Hansen[1], Valentina Barletta [1], Trine S. Dahl-Jensen[1], Anne Sofie Søndergaard [14], Beata M. Csatho [15], Ingo Sasgen [11], Jason Box [12] & Toni Schenk[15]

The Greenland Ice Sheet is the largest land ice contributor to sea level rise. This will continue in the future but at an uncertain rate and observational estimates are limited to the last few decades. Understanding the long-term glacier response to external forcing is key to improving projections. Here we use historical photographs to calculate ice loss from 1880–2012 for Jakobshavn, Helheim, and Kangerlussuaq glacier. We estimate ice loss corresponding to a sea level rise of 8.1 ± 1.1 millimetres from these three glaciers. Projections of mass loss for these glaciers, using the worst-case scenario, Representative Concentration Pathways 8.5, suggest a sea level contribution of 9.1–14.9 mm by 2100. RCP8.5 implies an additional global temperature increase of 3.7 °C by 2100, approximately four times larger than that which has taken place since 1880. We infer that projections forced by RCP8.5 underestimate glacier mass loss which could exceed this worst-case scenario.

[1] DTU Space, Technical University of Denmark, Kongens Lyngby, Denmark. [2] University of Copenhagen, Copenhagen, Denmark. [3] Bristol Glaciology Centre, University of Bristol, Bristol, UK. [4] Department of Earth System Science, University of California, Irvine, USA. [5] School of Earth Sciences, Ohio State University, Columbus, OH, USA. [6] Globe Institute, University of Copenhagen, Copenhagen, Denmark. [7] Institut des Géosciences de l'Environnement, Université Grenoble Alpes, Grenoble, France. [8] Center for global Sea Level Change, New York University Abu Dhabi, Abu Dhabi, UAE. [9] University of Alaska Fairbanks, College, AK, USA. [10] School of Geodesy and Geomatics, Wuhan University, Wuhan, China. [11] Glaciology Section, Alfred Wegener Institute, Bremerhaven, Germany. [12] Geological Survey of Denmark and Greenland, Copenhagen, Denmark. [13] Earth System Science Programme, The Chinese University of Hong Kong, Hong Kong, China. [14] Department for Geoscience, Aarhus University, Aarhus, Denmark. [15] Department of Geology, University at Buffalo, Buffalo, NY, USA. ✉email: abbas@space.dtu.dk

Sea level rise poses a serious threat to coastal areas worldwide. Global mean sea level (GMSL) rose by ~17 centimetres during the 20th century in response to the loss of land-based ice mass, thermal expansion of the oceans, and changes in terrestrial water storage[1–12]. This number could increase to 0.7–2 meters by 2100, mainly owing to accelerating ice loss[1]. During the past decade, the ice loss rate has been increasing and models project further acceleration over the coming decades[2–5]. Acceleration of ice discharge into the ocean is one of the primary drivers of mass loss and improving our understanding of how Greenland's outlet glaciers respond to external forcing is critical in order to reduce the uncertainty in future projections of mass loss[1]. In particular, very little is known about the centennial dynamic response of the Greenland Ice Sheet to atmosphere and ocean temperature variability[13].

The margin of the Greenland Ice Sheet has significantly changed since the end of the Little Ice Age and so looking at the response of the ice sheet over the past century provides an invaluable insight in how the ice discharge changes when climate warms[7,10]. Although the coasts of northwest and southeast Greenland are characterized by a large number of marine-terminating outlet glaciers with relatively small drainage areas[14], three outlet glaciers stand out owing to the size of their catchments. Jakobshavn Isbræ, Kangerlussuaq Glacier, and Helheim Glacier jointly drain ~12% of the Greenland Ice Sheet surface area, and hold enough ice to raise sea level by ~1.3 m[2,7]. Ice flow velocities in the Jakobshavn Isbræ and Kangerlussuaq Glacier region are increasing[15,16] and their glacier termini are retreating rapidly[17]. Jakobshavn Isbræ and Kangerlussuaq Glacier have a retrograde bed slope[18,19] (a bed that deepens inland) that lies below sea level, which makes them potentially susceptible to the Marine Ice Sheet Instability already observed in parts of West Antarctica[20]. In contrast, Helheim Glacier does not have a bed that deepens inland[18,21], allowing us to assess the importance of retrograde bed slopes by comparing Helheim Glacier with Jakobshavn Isbræ and Kangerlussuaq Glacier (see Fig. 1).

The importance of fjord and bed geometry as controls on the timing and magnitude of glacier retreat has been investigated in several studies[21–23]. Recent studies analysed ice front change of marine-terminating outlet glaciers in Greenland over the last 2–3 decades to show that higher retreat rates are associated with glaciers retreating into widening fjords or retrograde bedrock slope[22,23]. Here, we expand the observational record fourfold and focus on glacier retreat during 1880–2012 and the long-term impact of fjord geometry.

Although several studies have estimated the mass loss of Jakobshavn Isbræ, Kangerlussuaq Glacier, and Helheim Glacier during the satellite-era from the 1970s onwards at various spatial scales[15,16,24–26], estimates of decadal-scale temporal change during the 19th and 20th centuries remain poorly constrained and are limited to historical and geological records[27–30]. This limits our ability to assess the spatial and temporal extent of dynamic changes that followed the last short-lived glacier advance during the Little Ice Age. Integrating Jakobshavn Isbræ, Kangerlussuaq Glacier, and Helheim Glacier's long-term ice-dynamic memory in current projections, and understanding the relationship between driving mechanisms and climate variability, is key for providing accurate and robust estimates of present and future dynamic behaviour of these glaciers[1–5,31].

Existing estimates of ice mass loss over the 20th century only used a model of surface mass balance (SMB)[10] (the sum of snowfall minus melt) or aerial imagery[7]. However, the long-term records of ice front positions and surface lowering provide information on the dynamic behaviour of the outlet glaciers. In particular, a recent study provides evidence for a close relationship between frontal position and discharge[26]. A number of studies have used aerial stereo-photogrammetric imagery and reports from Greenland expeditions during the nineteenth and twentieth centuries to map ice front positions or ice surface lowering over the past century at Jakobshavn Isbræ[6,7,32], Kangerlussuaq Glacier[21,32,33] and Helheim Glacier,[21,32,33]. They used photogrammetric imagery acquired in 1944, 1953, 1959, and 1964[6,32]. For 1902, 1913, and 1933, they use estimates of Anker Weidick[34], which are based on an expedition by Lauge Koch in 1913 and the topographic mapping campaign of 1933[35]. For Kangerlussuaq Glacier and Helheim Glacier, glacier front retreat and ice surface lowering was obtained from historical photographs acquired in 1932, 1943, 1965, 1972, 1979, and 1981[21,32]. Although previous studies have focused on front retreat and ice surface lowering at a few selected points, here, we estimate high resolution basin-wide mass changes.

A previous estimate of ice mass loss over the 20th century based on aerial imagery[7] did not consider ice loss between the 1875 and the 2002 ice margin (this corresponds to the area between the dashed yellow curve "1875-front position" and the solid blue curve "2002-front position" in Fig. 2a) Although the previous study[7] provides a single century-scale ice loss estimate, here we estimate multi-year to decadal-scale ice losses, which are essential to understand how the ice dynamics of major drainage basins respond to variability in the atmosphere and ocean. We provide a century-long time series of ice mass change separated into ice dynamics and SMB components. In addition, we estimate the rebound of the underlying bedrock owing to ice mass loss, and local sea level lowering near the glacier termini, which makes it possible to investigate the importance of this potential negative feedback for the marine ice sheet instability hypothesis[20].

## Results

**Aerial stereo-photogrammetric imagery.** We use aerial stereo-photogrammetric imagery recorded in 1985 to map trimlines associated with the maximum extent of the Greenland Ice Sheet during the Little Ice Age, thereby quantifying vertical changes in ice surface elevation between then and 1985. We improve the approach described in Kjeldsen et al.[7] to estimate mass loss by including Little Ice Age maximum extent trimlines located in the area between the early 1900s and the 2002 ice margin. This allows us to estimate mass loss of the grounded ice that retreated to its 2002 position in the course of the 20th Century.

Frontal retreat and elevation changes are shown in Figs. 2 and 3 (see Supplementary Fig. 6 and Supplementary Movie 1). The left column in Fig. 1 shows orthophotos of Jakobshavn Isbræ, Kangerlussuaq Glacier and Helheim Glacier acquired in 2019 draped onto a Digital Elevation Model[7,21]. Orange lines show the extent during the Little Ice Age maximum. The temporal evolution of ice mass change and the associated sea level rise equivalent are shown in Fig. 4, Tables 1 and 2.

Elevation at Little Ice Age maximum extent are derived from direct observations of moraines and trimlines (Supplementary Figs. 7 and 8). Figure 3 shows reconstructions of surface elevation since 1875 constrained by moraines and trimlines shown in Fig. 1a. Changes in elevation are extrapolated to the ice sheet interior using a scale-value approach, with site-specific interpolations on a 0.5 × 0.5 km grid as described in Kjeldsen et al.[7]. However, for each 0.5 × 0.5 km grid point, we estimate scale values, one for each considered time interval. Figure 3 shows elevation and retreat in 1902, 1913, 1931, 1946, 1959, 1964, 1987, 2002, and 2012. Supplementary Movie 1 shows an animation of annual elevation change and retreat during 1875–2012. We use a linear interpolation of elevation and retreat to fill time intervals with no observations.

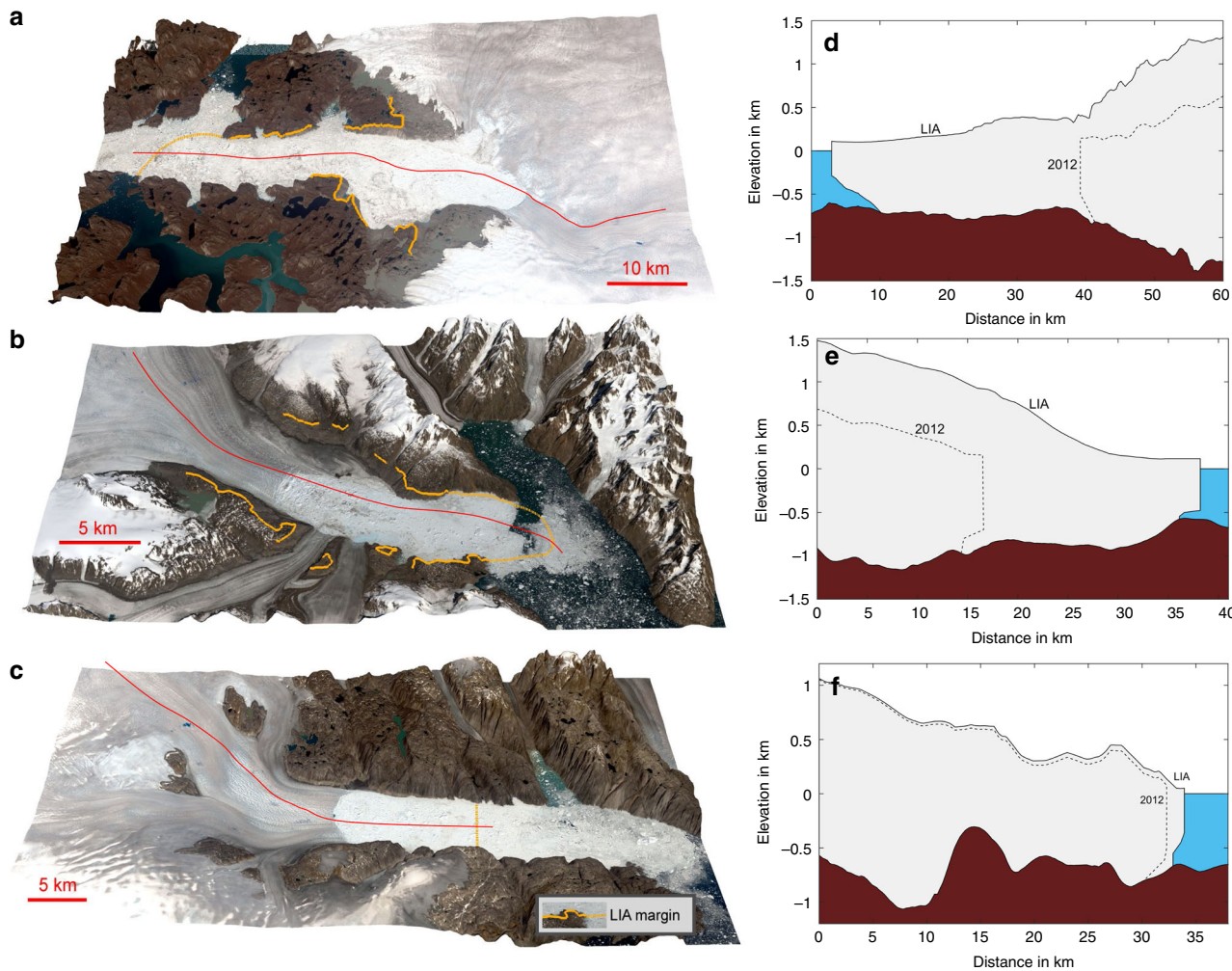

**Fig. 1 Little Ice Age margin and surface elevation. a** Landsat image of Jakobshavn Isbræ from 2019. Yellow line represents Little Ice Age (LIA) maximum extent. Red line denotes central flow line. **b** same as **a** for Kangerlussuaq Glacier. **c** same as **a** for Helheim Glacier. **d** Bed topography[18] and ice surface elevation during Little Ice Age maximum ice extent at Jakobshavn Isbræ. The profile follows the red line shown in **a**. **e** same as **d**, but for Kangerlussuaq Glacier. **f** same as **d**, but for Helheim Glacier.

To isolate dynamically induced ice mass changes, we use the Box reconstruction[10], hereafter BOX, to estimate SMB fluctuations. To obtain anomalies, we remove the 1961–1990 mean annual SMB. The SMB, equal to net snow accumulation minus snow and ice meltwater runoff, was reconstructed on a 5 km grid for Greenland ice from 1840 to 2012[10] and validated using data from the K-transect[36] along the western ice sheet, with an Root Mean Square Error of ~0.45 m water equivalent. The blue curves in Fig. 4 denote SMB induced ice mass changes.

**Ice mass loss over the 20th century.** We estimate a total mass loss of Jakobshavn Isbræ of 1518 ± 189 Gt over the period 1880–2012, with an increased rate in the early 1900s and 1930s associated with the collapse of its ice shelf (see Fig. 4). This implies a mean rate of ~13 Gt/yr over a century, with the dominant signal due to a combination of termination of the Little Ice Age and warming in the 1930s. Since the early 2000s, the rate of loss has been closer to 20–30 Gt/yr, consistent with previous estimates[15,16]. During 1900–2012 Kangerlussuaq Glacier lost a total of 1381 ± 178 Gt, with a mean rate of ~12 Gt/yr over a century. Owing to the lack of imagery during the past century, we cannot conclude whether the ice loss occurred in episodic events or at constant ice loss rate throughout the century. However, from the 1960s to late 1990s, both Jakobshavn Isbræ and

Kangerlussuaq Glacier showed a minor mass gain followed by a slight loss. Despite draining a huge area, we estimate a modest ice loss for Helheim Glacier of just 31 ± 21 Gt during 1900–2012. Our Fig. 4 and Table 1 suggest that changes in ice dynamics at Jakobshavn Isbræ and Kangerlussuaq Glacier are the major contributor to total ice loss, whereas ice loss at Helheim Glacier is equally distributed between ice dynamics and SMB.

Figure 2 shows the Little Ice Age maximum ice extent, along with 1980s and present-day frontal positions of Jakobshavn Isbræ, Kangerlussuaq Glacier, and Helheim Glacier. Although Jakobshavn Isbræ and Kangerlussuaq Glacier have retreated tens of kilometres since 1900 (Fig. 2a, b), Helheim Glacier has retreated and re-advanced, resulting in a net retreat in 2012 of only five kilometres with respect to 1900 (Fig. 2c). The Little Ice Age maximum extent of Helheim Glacier is situated in a place where the bed does not deepen inland. Although the bedrock does have a negative bed-slope 10 km further inland, a substantial bedrock peak between the present terminus location and the inland deepening presently prevents the glacier from retreating (Fig. 1f). In contrast, Jakobshavn Isbræ and Kangerlussuaq Glacier have retreated tens of kilometres since 1900 and lost ice corresponding to >8 mm of global sea level equivalent (Fig. 4a, b). At the Little Ice Age maximum extent, Kangerlussuaq Glacier's grounding line was situated on a bedrock bump (Fig. 1e).

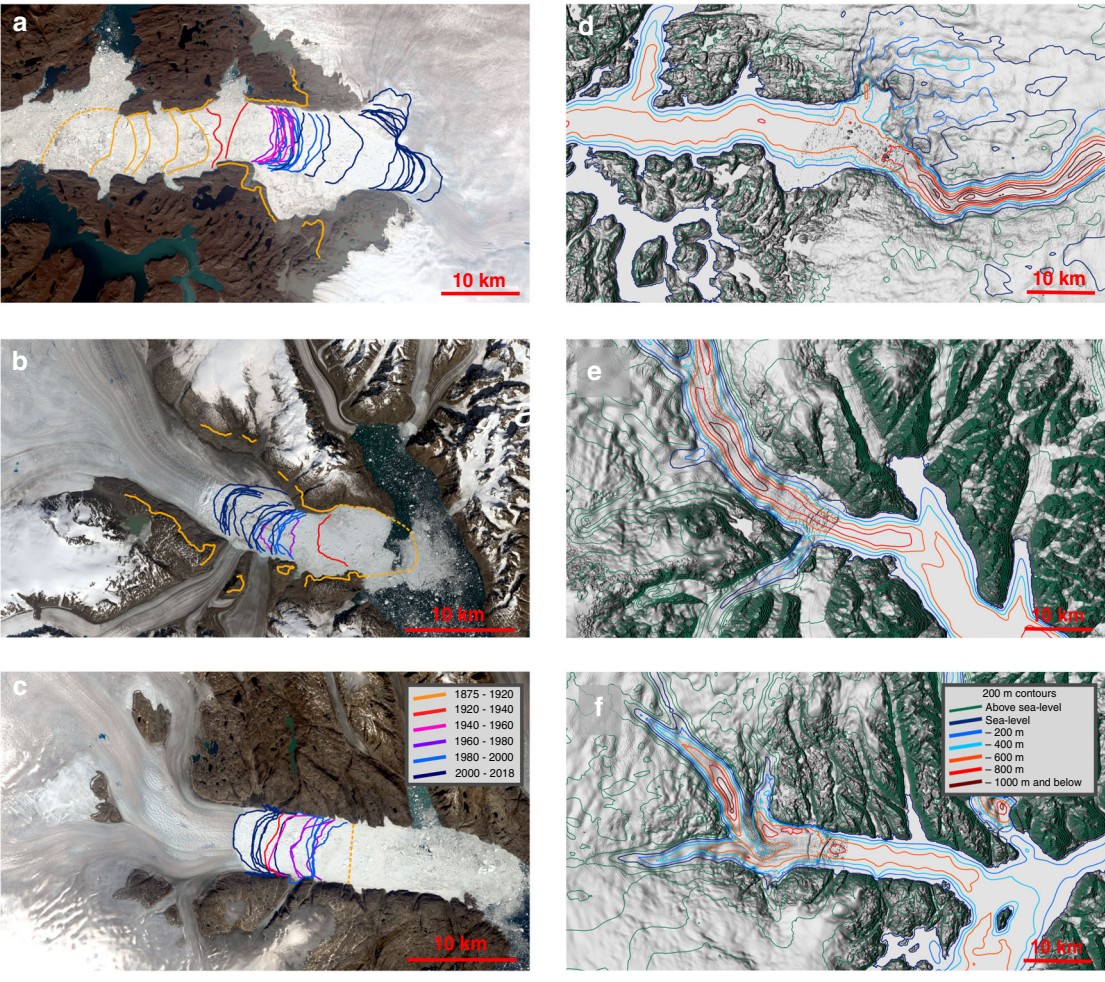

**Fig. 2 Frontal retreat and bed elevation. a** Frontal positions from 1875 to 2018 for Jakobshavn Isbræ. **b** Frontal positions from 1880 to 2018 for Kangerlussuaq Glacier. **c** Frontal positions from 1880 to 2018 for Helheim Glacier. **d** Bed elevation contour map of Jakobshavn Isbræ. **e** Bed elevation contour map of Kangerlussuaq Glacier. **f** Bed elevation contour map of Helheim Glacier.

Warmer ocean and air temperatures during the past century likely triggered a retreat[30], which resulted in >1000 Gt of ice loss by the mid-twentieth century (Fig. 4b). Kangerlussuaq Glacier experienced no significant ice loss during the cold period 1960–1970s[36] and entered a region with a slightly prograde bed slope at >20–30 km along profile in Fig. 1e. However, in recent decades, Kangerlussuaq Glacier has retreated into an area with retrograde slopes inland for tens of kilometres (Fig. 1e). During the past century, Jakobshavn Isbræ has retreated by ~40 km with major episodic retreats and mass loss in 1900s, 1930s, and 2000s likely triggered by a combination of atmosphere and ocean warming[37] (Figs. 2a and 4a). For example, warm waters may lead to a reduction of buttressing floating ice tongues, which may result in a positive feedback between retreat, thinning, and outlet glacier acceleration[26,38].

Our results clearly demonstrate that Jakobshavn Isbræ and Kangerlussuaq Glacier have gone through periods of dynamic instability (Figs. 2 and 3 and Supplementary Fig. 6) throughout the 20th century and are sensitive to small fluctuations in atmosphere and/or ocean warming[13,37–39]. Interestingly, rates of mass loss for Jakobshavn Isbræ in the early 1900s are comparable to present-day rates (Fig. 4a) and for Kangerlussuaq Glacier they were larger than present-day from 1880 to 1930. Both glaciers possess a retrograde bed slope at the present frontal position that persists for tens of kilometres inland (Fig. 1d, e). Furthermore, the 2012 frontal position of Jakobshavn Isbræ coincides with a steepening in bed slope, suggesting even greater sensitivity to dynamic instability[2–5]. As a consequence, both glaciers are likely to continue to retreat and lose mass[26]. Ice loss rates should increase since the bed slope steepens inland (Fig. 1d, e). Despite Helheim Glacier being remarkably stable over the past century (Fig. 4c), the glacier is now showing signs of retreat (Fig. 2c). In the summer of 2005 it entered a record-breaking retreat that came close to passing the bedrock peak located ~18 km inland from the Little Ice Age ice margin in Fig. 2c. The retreat was triggered by warm deep ocean water[39]. As this type of event will occur more frequently in the future as ocean warming continues, Helheim Glacier may pass its current pinning point (Fig. 1f) initiating a new phase of retreat and increased mass loss.

**Solid earth uplift and local sea level lowering.** Beside atmosphere and ocean cooling[40], a potential mechanism that can have a stabilizing effect on a retreating marine glacier is the solid Earth uplift and local sea level lowering as modelled and observed in Antarctica[41,42]. Over the 20th century climate has become warmer in the Arctic and as glaciers lose mass the pressure at the bedrock surface decreases resulting in uplift of the solid earth. The earth's instantaneous elastic response to contemporary changes in ice mass (Supplementary Fig. 1), and glacial isostatic adjustment i.e., the delayed viscoelastic response to past changes in ice loads (Supplementary Fig. 1) show meter-level land uplift

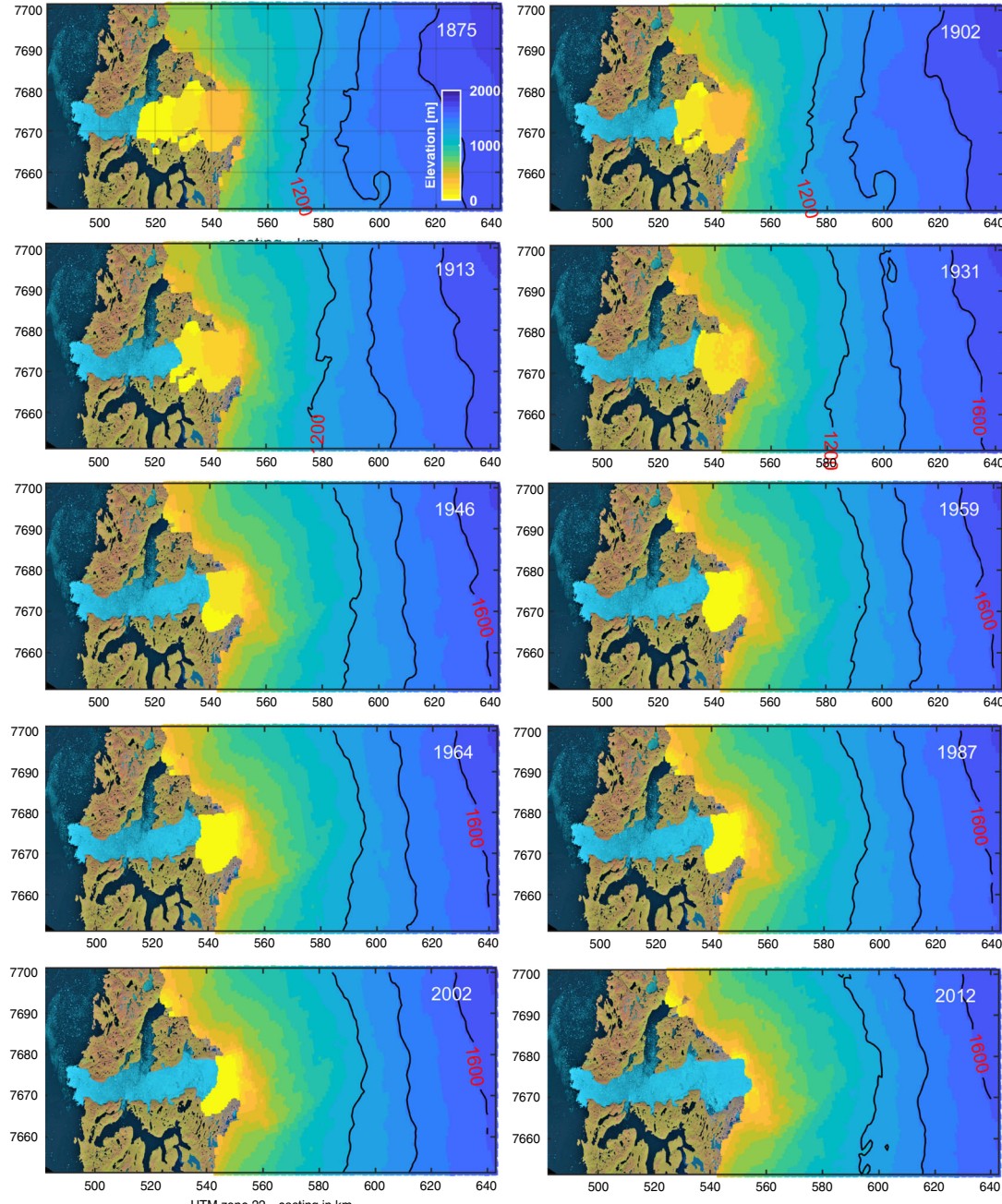

**Fig. 3 Retreat of Jakobshavn Isbræ.** Retreat and surface elevation in meter of Jakobshavn Isbræ in 1875, 1902, 1913, 1931, 1946, 1959, 1964, 1987, 2002, and 2012.

over the 20th century. Decreases in local gravity caused absolute sea level lowering shown in Supplementary Fig. 1. In total, the relative sea level lowered by ~280 cm and 350 cm during 1880–2012 near the present-day grounding lines of Jakobshavn Isbræ and Kangerlussuaq Glacier, respectively (Fig. 5a, b). For Helheim Glacier we estimate a much smaller relative sea level lowering of ~40 cm near its grounding line (Fig. 5c). We posit that the change in local sea level has not had a strong effect on ice dynamics, as these glaciers are tidewater glaciers and do not have a significant floating section. Sea level fall therefore does not lead to a grounding line advance and only changes the water pressure at the calving front. The negative feedback observed and modelled in Antarctica[41,42] is therefore not a relevant mechanism in Greenland and we do not expect it to have any significant influence on ice dynamics and ice front retreat.

## Discussion

We have presented ice mass change estimates with significantly improved temporal resolution during the 20th century (Fig. 4), which is essential to understanding long-term glacier dynamics and its relation to climate forcings[1–5]. We take into account ice lost during the retreat from the Little Ice Age maximum extent and its 2012 position. Our data improvement show that neither uplift of bedrock nor sea level lowering[41,42] (Fig. 5 and Supplementary Fig. 1) owing to a decrease in local gravity have had a major stabilizing effect on glacier retreat and ice mass loss as Greenland's outlet glaciers are tidewater glaciers. As both Jakobshavn Isbræ and Kangerlussuaq Glacier retreat towards deeper and steeper beds[18,19] (Fig. 2), sea level lowering will not prevent drawdown of ice mass loss in a warming climate over the next centuries.

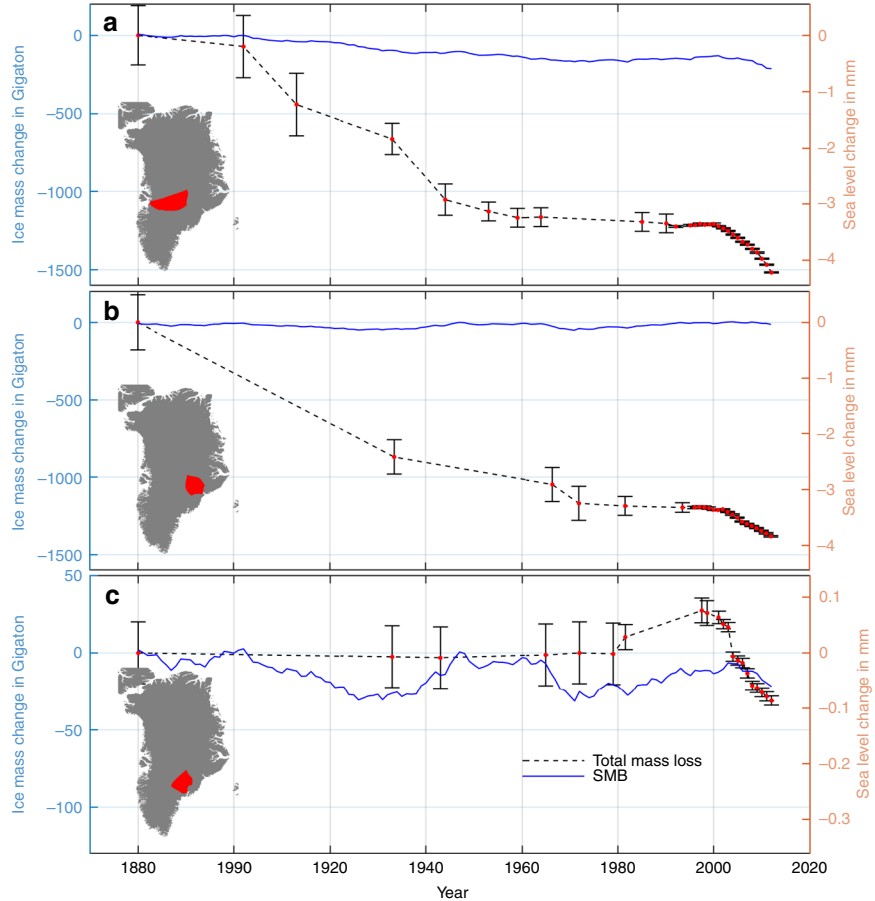

**Fig. 4 Ice mass change.** Time series of accumulated ice mass change from 1880 to 2012 for **a** Jakobshavn Isbræ. **b** Kangerlussuaq Glacier. **c** Helheim Glacier. Negative values denote mass loss. Black dashed curve denotes total observed ice mass loss change and the vertical bars denote uncertainty. The blue curve denotes SMB. Map of Greenland is shown in lower left corner and the red area denotes drainage area of the considered glacier. Right axis shows sea level change in mm and the left axis shows the corresponding ice mass change in gigaton.

**Table 1 Basin-wide mass loss of Jakobshavn Isbræ, Kangerlussuaq Glacier, and Helheim Glacier.**

| Basin | Cumulative mass loss | | | | | Sea level equivalent |
|---|---|---|---|---|---|---|
| | Gigaton | | | | | mm |
| | 1875–1932 | 1932–1964 | 1964–1981/5 | 1981–2012 | 1875–2012 | 1875–2012 |
| Jakobshavn Isbræ | 661 ± 199 | 503 ± 199 | 91 ± 199 | 322 ± 60 | 1518 ± 189 | 4.2 ± 0.5 |
| Kangerlussuaq Glacier | 870 ± 205 | 178 ± 205 | 138 ± 205 | 195 ± 30 | 1381 ± 178 | 3.8 ± 0.5 |
| Helheim Glacier | −3 ± 20 | −4 ± 20 | −9 ± 20 | 44 ± 8 | 31 ± 21 | 0.1 ± 0.1 |
| Total mass loss | 1528 ± 424 | 677 ± 424 | 220 ± 424 | 561 ± 98 | 2930 ± 322 | 8.1 ± 0.9 |
| Percentage dynamic ice loss | 89% | 96% | 77% | 92% | 91% | 91% |

**Table 2 Basin-wide mass loss rate of Jakobshavn Isbræ, Kangerlussuaq Glacier, and Helheim Glacier.**

| Basin | Mass loss rate in Gigaton/year | | | | |
|---|---|---|---|---|---|
| | 1875–1932 | 1932–1964 | 1964–1981/5 | 1981–2012 | 1875–2012 |
| Jakobshavn Isbræ | 13 ± 4 | 16 ± 6 | 4 ± 9 | 12 ± 2 | 11 ± 1 |
| Kangerlussuaq Glacier | 17 ± 4 | 6 ± 6 | 5 ± 9 | 6 ± 1 | 10 ± 1 |
| Helheim Glacier | 0 ± 1 | 0 ± 1 | 0 ± 1 | 1 ± 0.3 | 0 ± 1 |

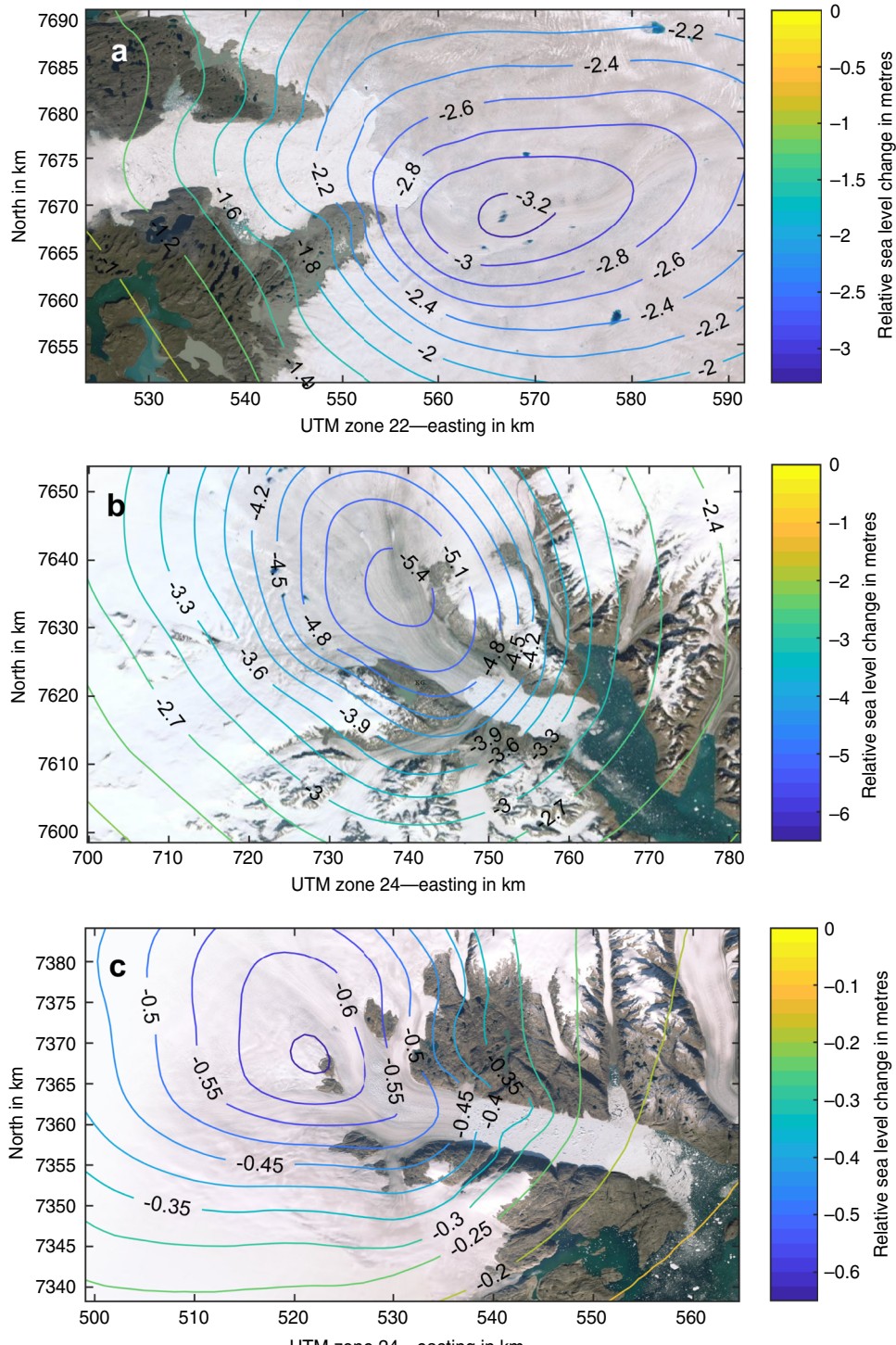

**Fig. 5 Relative local sea level change.** Relative local sea level change during 1880–2012 in metres for **a** Jakobshavn Isbræ (top), **b** Kangerlussuaq Glacier, **c** Helheim Glacier caused by combined effect of elastic uplift, sea level change due to reduced gravity and uplift due to glacial isostatic adjustment.

To estimate ice loss during the 21st century models typically use four greenhouse gas emission scenarios[2–4], RCP2.6, RCP4.5, RCP6, and RCP8.5, where the latter is considered as the worst-case scenario. RCP8.5 corresponds to a global mean temperature rise of 3.7 ± 0.7 °C above the 1986–2005 reference period[43], which equates to ~8.3 ± 1.9 °C over Greenland, accounting for polar amplification[43,44]. Using RCP8.5 as external forcing[43,42], an ice flow model suggests that the ice loss of Jakobshavn Isbræ, Helheim Glacier and Kangerlussuaq Glacier could contribute

9.1–14.9 mm to sea level rise by 2100[4]. However, during the 20th century the average air temperature over Greenland rose[10] by only 1.5 °C and led to a sea level rise of 8.1 ± 1.1 mm from those three glaciers (see Table 1). All three have retreated into regions with widening fjords and two into areas of retrograde bedrock slope[21–23]. Combined with the approximately five times larger temperature increase predicted by 2100 for RCP8.5 compared with that seen since the end of the Little Ice Age[10], it suggests that the model projections underestimate the worst-case mass loss

from these three glaciers. It also seems likely that this is not limited to just these three glaciers. Attribution of the cause for the possible underestimation is beyond the scope of this study but multiple lines of evidence suggest that many deterministic ice sheet models may underestimate the sensitivity of the ice sheet to external forcing[1,43].

The centennial dynamic responses of Jakobshavn Isbræ, Kangerlussuaq Glacier, and Helheim Glacier account for ~90% of their total mass loss (see Table 1), highlighting the importance of understanding those glacier's variability and long-term dynamic response to external forcing[13,23]. Our findings suggest that, whereas local bed geometry is an important control on glacier stability[22,23,42,43], changes in atmospheric and oceanic forcings[13,37–40] can lead to rapid and extensive retreat that need to be captured by numerical models, as they are the primary driver of mass loss. These long-term observations provide strong constraints on past glacier variability that should be used to validate models in order to increase the reliability of future projections.

## Methods

**Negative feedback**. We estimate the local sea level change as a sum of (i) elastic uplift due to ice loss, (ii) sea level change due to reduced gravity, and (iii) uplift due to glacial isostatic adjustment. (i–ii) To estimate elastic uplift and sea level change owing to reduced gravity, we convolve mass loss estimates during the Little Ice Age max and 2012 with the Green's functions derived by Wang et al.[45] for elastic Earth model iasp91 with refined crustal structure from Crust 2.0[45]. (iii) GIA: We adopt the GNET-GIA model of Khan et al.[46]. The total effect of all three components is shown in Fig. 5.

Supplementary Fig. 1 shows the elastic uplift, sea level change due to reduced gravity, and uplift due to glacial isostatic adjustment at Jakobshavn Isbræ, Kangerlussuaq Glacier and Helheim Glacier. The associated uncertainties are shown in Supplementary Fig. 2.

**Correlation between retreat and elevation changes**. Several studies have shown linear correlation between glacier surface speed, changes in front position, and changes in surface elevation. Bevan et al.[47] and Khan et al.[21] used data from 1985 to 2012 to show long-term correlation between thinning/thickening and retreat/re-advance at Kangerlussuaq Glacier and Helheim Glacier. Supplementary Figs. 4 and 5 show time series of elevation change and retreat at two different positions located at Jakobshavn Isbræ. Elevation time series at Point A was adopted from Csatho et al.[6]. The glacier retreated beyond this point in 2001. Point B is located farther upstream at about 865 m elevation. This position is selected because it was measured by multiple ATM campaigns[48] starting in 1994. The 1985-elevation is based on aerial stereo-photogrammetric imagery[29].

Supplementary Fig. 6 shows the drainage area of Jakobshavn Isbræ, Kangerlussuaq Glacier, and Helheim Glacier. In total, the three glaciers drain ~12% of the Greenland Ice Sheet. Figure 3 shows ice surface elevations in meters of Jakobshavn Isbræ during 1875, 1902, 1913, 1931, 1946, 1964, 1987, 2002, and 2012. We use the geodetic approach as described by Kjeldsen et al.[7] (see their Methods section) to calculate spatially distributed ice thinning patterns and basin-wide mass balance of the Jakobshavn Isbræ, Kangerlussuaq Glacier and Helheim Glacier. We calculate elevation change grid with spatial resolution of 500 × 500 meters. Supplementary Movie 1 and Supplementary Fig. 6 shows temporal evolution of retreat and surface lowering of Jakobshavn Isbræ and Kangerlussuaq Glacier, respectively. Time series of total mass loss of Jakobshavn Isbræ is shown in Fig. 2.

The photos from 1944, 1953, 1959, 1964, 1985 have elevation RMS errors of 2–4 m. The photos from 1902, 1913, 1933 have an RMS of 25 m. Error estimations are described in details in Csatho et al.[6]. Trimlines (Supplementary Figs. 7 and 8) have a height error of 10 m and are likewise described in Csatho et al.[6].

**Error assessment using 1978–2012 discharge and SMB**. Our estimates of mass balance are consistent with recent independent estimates of mass balance of Jakobshavn Isbræ, Kangerlussuaq Glacier, and Helheim Glacier derived from surveys of thickness, surface elevation, velocity, and SMB from 1972 to 2018[15] (Supplementary Fig. 9). In a recent study, Mouginot et al.[15] reconstruct the mass balance of Jakobshavn Isbræ, Kangerlussuaq Glacier, Helheim Glacier and other glaciers using a comprehensive survey of thickness, surface elevation, velocity, and SMB from 1972 to 2018. The green curves in Supplementary Fig. 9 show basin-wide mass loss estimates from Mouginot et al.[15], whereas the black curve denotes mass loss and associated error estimates of this study based the method described by Kjeldsen et al.[7]. In general, mass loss estimates from the two studies are consistent and agree within the error bars. The mass gain of Helheim Glacier after 1980 is consistent with observed thickening of ~50 meters[21].

**Error of digital elevation models**. We use data validated and published by Csatho at al.[6], Kjeldsen at al.[7], Khan et al.[21], Korsgaard et al.[28], Schenk et al.[32]. Supplementary Fig. 10 shows an example of hillshade of the 1964 digital elevation model. On 21 June 1964, an aerial survey was conducted of Jakobshavn Isbræ in connection with the Expédition Glaciologique Internationale Au Groenland[49,50]. We use the aerial photos from strip 272 C scanned on a photogrammetric scanner and the calibration report. Coordinates and heights for the 21 ground control points were obtained from the pilot mapping project conducted by the Danish Agency for Data Supply and Efficiency in the Disko Bugt region. These are curated and edited Pleiades DEM and orthophotos and are available for download here: https://download.kortforsyningen.dk/content/geodataprodukter?field_korttype_tid_1=661.

Local heights in Greenland Vertical Reference GVR2016 (EPSG: 8267) have been transformed into GR96 ellipsoidal heights (EPSG:4909) (https://github.com/OSGeo/proj-datumgrid/tree/master/north-america). The weight-normalized bundle adjustment gives rms values as follows: $rms_x$ and $rms_y$ are 1.6 m, and $rms_z$ is 3.5 m.

## Data availability

The data sets generated during and/or analysed during the current study are available in the DTU repository, [https://ftp.space.dtu.dk/pub/abbas/naturecomm2020/].

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

## Acknowledgements

S.A.K. acknowledge support from the Danish Council for Independent Research. J.L.B. acknowledges support from European Research Council grant number 694188 (GlobalMass). J.M. acknowledges support from the French National Research Agency (ANR) grant (ANR-19-CE01-0011-01). Z.B. acknowledges support from the China Postdoctoral Science Foundation (2019T120687) and the Fundamental Research Funds for the Central Universities (2042020kf0009). I.S. acknowledges funding by the Helmholtz Climate Initiative REKLIM (Regional Climate Change), a joint research project of the Helmholtz Association of German Research Centres (HGF). A.A. acknowledge support from NASA grant NNX17AG65G and NSF grant PLR-1603799. D.M.H. acknowledges support from grant G1204 of NYU Abu Dhabi's Center for global Sea Level Change and NSF grant ARC-1304137. N.K.L. acknowledges support from Villum Foundation (grant no. 023440). A.A.B. was funded by the Carlsberg Foundation (Grant CF17-0529). W.C. and A.L. acknowledges support from Danish Council for Independent Research (grant no. 8049-00003B). B.M.C. acknowledges support from NASA's Operation IceBridge and Sea Level Change science team grants, NNX17AI65G and 80NSSC17K0611, respectively. We thank professor Hansheng Wang for providing Load Love numbers and Green's functions.

## Author contributions

S.A.K. conceived the study, analyzed parts of the data, especially glacier altimetry, bedrock uplift, basin mass loss, and prepared most of the figures and wrote parts of the paper. J.L.B. and M.M. wrote parts of the paper and delivered bedrock topography. A.A.B. prepared some of the figures and wrote parts of the paper. B.M.C., T.S., N.J.K., and A.A.B. created elevations from trimlines and aerial photos. J.Box analysed SMB data. M.B., K.H.K., J.M., A.L., D.M.H., A.A., B.Z., V.H., W.C., N.K.L., L.L., K.H., V.B., T.S.D.J., A.S.S., and I.S. helped in discussion and wrote parts of the paper.

## Competing interests

The authors declare no competing interests.
