## [Peer Review File · Nature Communications]

REVIEWER COMMENTS

Reviewer #1 (Remarks to the Author):

Review of “Centennial dynamic response of Greenland’s three largest outlet glaciers” by Khan et al.

Manuscript # NCOMMS-20-02-12911-T

This manuscript presents an analysis of centennial dynamic response of Greenland’s three largest outlet glaciers, Jakobshavn, Helheim, and Kangerdlugssuaq based on historical photographs. The analysis sounds solid and results are interesting, and the study helps address an important topic, how Greenland the largest land ice contributor to sea level rise responded to climate change at centennial time scales. I can recommend it be accepted for publication in Nature Communications after some major improvements.

Comments:

- 1) The presentation needs substantial improvements. This is a rather comprehensive analysis, and the authors are trying a little too hard to fit it into a Nature Communications article, which apparently affects the presentation of the results.
- 2) The authors packed too much content into Figure 1 (with 9 panels), making it too difficult to read. It is hard to imagine how people could read the contents clearly when in print format. The description in Figure 1 caption is not consistent with the displayed information. There are 9 panels in Figure 1, but the caption has described 10.
- 3) This manuscript appears to be “hastily” put together, without a careful proof-reading. There were some typos here and there (e.g., Ln 28: “three largest contributor”, Ln. 55 & 99: LIA was defined twice, Ln. 104: “Box reconstruction”, ... etc.).
- 4) The Method section relies too heavily on figures in Extended Data, which greatly affects the readability of the paper without frequently switching between the main paper and Extended Data. Some of the contents in Method are more suitable for the main paper. The number of illustrations (11 in total) in Extended Data also exceeds the journal limitation.

Other Minor Comments:

- 5) Ln 116: Using two citations (15, 16) here is misleading. As it states, the authors simply present their own estimates of JI mass rates here (“Since the early 2000’s, the rate of loss has been closer to 20-30 Gt/yr”). To make more sense, the authors might want to add “consistent with previous estimates” followed by the citations.
- 6) Ln 130-131: The authors claim that “HG has maintained its position”. This statement is a little arbitrary. HG is certainly more stable than the other two glaciers, it has still retreated over 5 km (~ 1/3 of KG’s) over the studied period.

- 7) Ln 183-186: The last sentence has stopped short of stating an important conclusion (mentioned in in the summary, “projections may be underestimated and glacier mass loss could exceed worst-case scenario”).
- 8) Authors’ contribution information is missing.

Reviewer #2 (Remarks to the Author):

Review of NCOMMS-20-12911-T

"Centennial dynamic response of Greenland's three largest outlet glaciers"

Interest in using historical photographs for assessing long term glacier change has received considerable attention. For some scientists developments in computer-vision has rejuvenated interest in the use of historical photographs for glaciological investigations. However, this paper applies an aerial stereo photogrammetric approach adapted from by Kjeldsen et al. (2015) to investigate mass loss in Greenland. The submitted manuscript delivers on this by extracting useful glaciological information from additional image-sets, thus builds on existing work by providing a multi-temporal insight into the response of three marine terminating glaciers draining 12% of the Greenland Ice sheet (GrIS). This novelty is communicated in the abstract and section on 'Aerial stereo photographic imagery' (Lines 72 - 102) and the study uses both photographs from the pre-satellite era alongside modern airborne and satellite derived data.

In lines 63-70 the key rationale and novelty of this particular study is outlined. It would be beneficial for the reader if the limits to existing studies looking at mass loss in the pre-satellite era could be briefly summarised. At present studies looking at change in the satellite era are clearly outlined to the reader (e.g., via citing references 15, 16, 21 and 22), but details on your existing studies that use historical photographs to investigate changes to the GrIS are, in my opinion, not adequately synthesised. This context would greatly enhance this part of the manuscript, and help highlight the value of the work undertaken. Some details are added in the proceeding section, so it is most likely a case of adding detail and restructuring what is already provided for clarity.

Some of the methods used are detailed in the section starting on line 72. The generation of usable data for monitoring glacier change is clearly important, so I can see why details are provided in the main body, however, this compromises the methods sections starting on 297, and vital information required is split between the sections. In lines 95 and 96 the authors highlight that modern airborne and satellite altimetry data is used for the 1985-2010 monitoring period. With this being the case, I am interested to note that total mass loss for the three sites (KG, JI, and HG) is reported up to 2012 (e.g. please see lines 112-123, Table 1, extended data Fig. 7 & 8). A discussion of the estimations of glacier retreat to the 2002 position are presented on lines 97 - 90 with reference to the LIA maximum extent trimlines. My comment related to this is that the multi-temporal data used here could be more clearly introduced. Can authors clarify regarding these dates and associated data sources? With the previous comment in mind, a table in the extended data detailing the various data sources used, resolution, associated error, would be a valuable — perhaps essential — addition to this manuscript (e.g. similar to table 1 in Kjær et al. 2012, but with more detail). Overall, whilst reference to the methods adopted in existing work (e.g. you cite Kjeldsen et al. 2015) does permit reproducibility, I would certainly welcome the inclusion of more information in the manuscript, specifically as the extraction of usable glaciological datasets from historical image-sets is not always a straight-forward task.

Given the importance of retrograde — or reverse bedrock slopes — as argued in the article, more information on the bathymetry of these basins could be provided. Could the authors consider displaying these data (perhaps as contours) on Figure 2 in panels a, b, and c? Are these undulations typically transverse to ice-flow and present across the entire fjord bottom? I certainly believe some morphometric description of the (former) glacier bed is warranted. At present, for example, I find the description of bedrock obstacles, described colloquially as 'bumps' inadequate. The manuscript sets the importance of glacier-specific factors up in the introduction (line 60-61), but does not adequately explore this later in the results and discussion. If the key 'story' is to be centred around this, then it must be discussed more rigorously in the latter sections of the manuscript. The authors should improve this section as a priority, and I am concerned that existing work is omitted from the manuscript. For example, the authors would benefit from

reviewing the work of Bunce et al. (2018) where the correlation between the presence of reverse bedrock slopes and retreat rate is clearly established through the analysis of a much larger dataset of marine terminating glaciers ($n = 276$). Why is this clearly relevant work not cited? How does your work advance understanding in relation to this despite the focus on three sites? Why is fjord geometry not also discussed? Indeed, in your earlier published work you highlight that "Helheim Glacier has retreated and advanced several times throughout the last century; however, likely due to rapid increase of the glacier width and bed elevation farther inland, the retreats have been followed by advances (e.g. during 2005–2006), suggesting both bed and width (see Fig. 11c) play an important role in the Helheim Glacier stability and long-term mass balance" (Khan et al., 2014: 1505). Should your manuscript more clearly establish what is already known about the dynamics of the glaciers studied, to allow you to more clearly communicate the novelty of the work presented here?

Moving on, the implications section clearly outlines that mass loss related to climatic forcing is not compensated by stabilising effects associated with changes in sea level and bedrock uplift. The implications of this are briefly reviewed. In the abstract it is argued that "projections may be underestimated and glacier mass loss could exceed worst-case scenario." The manuscript could be improved by returning to this point in the main body to elaborate in further detail. The details provided do not go significantly beyond the brief summary provided in the abstract.

Overall, this manuscript could be published in Nature Communications, however I believe it should first be enhanced with some refinements to the structure, figures, and discussion to sharpen the focus of the paper. I hope the authors find these constructive comments helpful for improving their manuscript. Below I also append a series of more minor comments on this draft of the manuscript – you may benefit from carefully proof-reading the manuscript to ensure the final draft is ready for publication.

Some specific comments:

Line 26: Responds - change to response?

Line 30: Suggestion — rephrase the list of glaciers and split this sentence.

Line 56: Provide a supporting reference.

Line 60: Provide a reference to support this (RE: the lack of retrograde bed slope at HG). - ref 18, 27?

Line 127: It is unclear why 'Extended Data Fig. 9' has been cited here.

Line 179: Rephrase this sentence.

Line 155: 'Bump' is rather colloquial.

Line 300: Arctic not artic.

Line 327: 'i.e.' — Not required?

Figure 1 — 'Assumed' ice margin? How was this 'assumed'? Please clarify.

Figure 1 — Please consider adding the source of bed topography here. Are these undulations typically transverse to ice-flow? Note my point regarding this above.

Extended Data Fig. 11 — Caption. Aerial not arial.

References cited

Bunce, C., Carr, J.R., Nienow, P.W., Ross, N. and Killick, R., 2018. Ice front change of marine-terminating outlet glaciers in northwest and southeast Greenland during the 21st century. *Journal of Glaciology*, 64(246), 523-535.

Khan, S. A., Kjeldsen, K. K., Kjær, K. H., Bevan, S., Luckman, A., Aschwanden, A., Bjørk, A. A., Korsgaard, N. J., Box, J. E., van den Broeke, M. R., van Dam, T. M., & Fitzner, A., 2014. Glacier dynamics at Helheim and Kangerdlugssuaq glaciers, southeast Greenland, since the Little Ice Age. *Cryosphere*, 8, 1497-1507.

Kjær, K.H., Khan, S.A., Korsgaard, N.J., Wahr, J., Bamber, J.L., Hurkmans, R., van den Broeke, M., Timm, L.H., Kjeldsen, K.K., Bjørk, A.A. and Larsen, N.K., 2012. Aerial photographs reveal late-20th-century dynamic ice loss in northwestern Greenland. *Science*, 337(6094), 569-573.

Kjeldsen, K.K., Korsgaard, N.J., Bjørk, A.A., Khan, S.A., Box, J.E., Funder, S., Larsen, N.K., Bamber, J.L., Colgan, W., Van Den Broeke, M. and Siggaard-Andersen, M.L., 2015. Spatial and temporal distribution of mass loss from the Greenland Ice Sheet since AD 1900. *Nature*, 528(7582), 396-400.

REVIEWER #1 comments and *our response*

Review of “Centennial dynamic response of Greenland’s three largest outlet glaciers” by Khan et al. Manuscript # NCOMMS-20-02-12911-T

This manuscript presents an analysis of centennial dynamic response of Greenland’s three largest outlet glaciers, Jakobshavn, Helheim, and Kangerdlugssuaq based on historical photographs. The analysis sounds solid and results are interesting, and the study helps address an important topic, how Greenland the largest land ice contributor to sea level rise responded to climate change at centennial time scales. I can recommend it be accepted for publication in Nature Communications after some major improvements.

Response: Thank you very much for reviewing this paper. In the light of insightful response from you and reviewer #2, we have made several changes, which have greatly improved the revised manuscript. A detailed response to your comments addressing all of the identified issues is listed below.

Comments:

1) The presentation needs substantial improvements. This is a rather comprehensive analysis, and the authors are trying a little too hard to fit it into a Nature Communications article, which apparently affects the presentation of the results.

Response: we have changed the format of the manuscript. We have added new figures and moved figures from “extended data” to the main text. We have added new paragraphs that explain the analysis. The new format fits Nature Communications much better.

2) The authors packed too much content into Figure 1 (with 9 panels), making it too difficult to read. It is hard to imagine how people could read the contents clearly when in print format. The description in Figure 1 caption is not consistent with the displayed information. There are 9 panels in Figure 1, but the caption has described 10.

Response: we agree, we have decided to split old figure 1 into two figures. Figure 1 now contains 6 panel. See new figure 1 and 2.

3) This manuscript appears to be “hastily” put together, without a careful proofreading. There were some typos here and there (e.g., Ln 28: “three largest contributor”, Ln. 55 & 99: LIA was defined twice, Ln. 104: “Box reconstruction”, ... etc.).

Response: Thanks. We have carefully updated the manuscript and corrected all typos.

4) The Method section relies too heavily on figures in Extended Data, which greatly affects the readability of the paper without frequently switching between the main paper and Extended Data. Some of the contents in Method are more suitable for the main paper. The number of illustrations (11 in total) in Extended Data also exceeds the journal limitation.

Response: We fully agree with the reviewer and have moved main results from Method to the main text.

Other Minor Comments:

5) Ln 116: Using two citations (15, 16) here is misleading. As it states, the authors simply present their own estimates of JI mass rates here (“Since the early 2000’s, the rate of loss has been closer to 20-30 Gt/yr”). To make more sense, the authors might want to add “consistent with previous estimates” followed by the citations.

Response: Corrected.

6) Ln 130-131: The authors claim that “HG has maintained its position”. This statement is a little arbitrary. HG is certainly more stable than the other two glaciers, it has still retreated over 5 km (~ 1/3 of KG’s) over the studied period.

Response: we agree and have changed the text to “HG has retreated and re-advanced resulting in a net retreat in 2012 of only few kilometres compared to 1900”.

7) Ln 183-186: The last sentence has stopped short of stating an important conclusion (mentioned in the summary, “projections may be underestimated and glacier mass loss could exceed worst-case scenario”).

Response: we have changed the entire section about implications. See new section entitled “Discussion”.

8) Authors’ contribution information is missing.

Response: Authors’ contribution information is now added

Reviewer #2 (Remarks to the Author):

Review of NCOMMS-20-12911-T

"Centennial dynamic response of Greenland’s three largest outlet glaciers"

Interest in using historical photographs for assessing long term glacier change has received considerable attention. For some scientists developments in computer-vision has rejuvenated interest in the use of historical photographs for glaciological investigations. However, this paper applies an aerial stereo photogrammetric approach adapted from by Kjeldsen et al. (2015) to investigate mass loss in Greenland. The submitted manuscript delivers on this by extracting useful glaciological information from additional image-sets, thus builds on existing work by providing a multi-temporal insight into the response of three marine terminating glaciers draining 12% of the Greenland Ice Sheet (GIS). This novelty is communicated in the abstract and section on ‘Aerial stereo photographic imagery’ (Lines 72 - 102) and the study uses both photographs from the pre-satellite era alongside modern airborne and satellite derived data.

Response: Thank you very much for reviewing this paper. Following the suggestions and comments from you and reviewer #1, we have made several changes, which have greatly improved the manuscript. A detailed response to your comments addressing all of the identified issues is listed below.

In lines 63-70 the key rationale and novelty of this particular study is outlined. It would be beneficial for the reader if the limits to existing studies looking at mass loss in the pre-satellite era could be briefly summarised. At present studies looking at change in the satellite era are clearly outlined to the reader (e.g., via citing references 15, 16, 21 and 22), but details on your existing studies that use historical photographs to investigate changes to the GrIS are, in my opinion, not adequately synthesised. This context would greatly enhance this part of the manuscript, and help highlight the value of the work undertaken. Some details are added in the proceeding section, so it is most likely a case of adding detail and restructuring what is already provided for clarity.

Response: Done. We have added a paragraph after lines 63-70 that briefly summarize literature that use historical photographs to investigate changes to the GrIS. We have also listed years when historical photographs were acquired.

Some of the methods used are detailed in the section starting on line 72. The generation of usable data for monitoring glacier change is clearly important, so I can see why details are provided in the main body, however, this compromises the methods sections starting on 297, and vital information required is split between the sections. In lines 95 and 96 the authors highlight that modern airborne and satellite altimetry data is used for the 1985-2010 monitoring period. With this being the case, I am interested to note that total mass loss for the three sites (KG, JI, and HG) is reported up to 2012 (e.g. please see lines 112-123, Table 1, extended data Fig. 7 & 8). A discussion of the estimations of glacier retreat to the 2002 position are presented on lines 97 - 90 with reference to the LIA maximum extent trimlines. My comment related to this is that the multi-temporal data used here could be more clearly introduced. Can authors clarify regarding these dates and associated data sources? With the previous comment in mind, a table in the extended data detailing the various data sources used, resolution, associated error, would be a valuable — perhaps essential — addition to this manuscript (e.g. similar to table 1 in Kjær et al. 2012, but with more detail). Overall, whilst reference to the methods adopted in existing work (e.g. you cite Kjeldsen et al. 2015) does permit reproducibility, I would certainly welcome the inclusion of more information in the manuscript, specifically as the extraction of usable glaciological datasets from historical image-sets is not always a straight-forward task.

Response: Thanks for finding the inconsistency on line 96. The monitoring period is 1985-2012 and not 1985-2010 as stated in the text.

We have moved old version of Extended data figure 7 (Jakobshavn surface elevations in meter during 1875-2012) to the main text and briefly describe the method. The method is described in details by Kjeldens et al (2015), see their Method section. The main difference between Kjeldens et al (2015) and this study is that we estimate several scale grid parameters on a 1x1 km grid, one for each considered time interval. While Kjeldsen consider a single grid for the entire time span from LIA_{MAX} to 1981.

Details on data source, or a reference to data source, in particular historical data from 1880-1980s is described in Csatho et al 2008 in Journal of glaciology (table 1 and 4) for Jakobshavn Isbræ and Schenk et al 2014 in Remote Sensing of Environment for Kangerdlugssuaq and Helheim glacier. Both papers are cited.

Given the importance of retrograde — or reverse bedrock slopes — as argued in the article, more information on the bathymetry of these basins could be provided. Could the authors consider displaying these data (perhaps as contours) on Figure 2 in panels a, b, and c? Are these undulations

typically transverse to ice-flow and present across the entire fjord bottom? I certainly believe some morphometric description of the (former) glacier bed is warranted. At present, for example, I find the description of bedrock obstacles, described colloquially as ‘bumps’ inadequate. The manuscript sets the importance of glacier-specific factors up in the introduction (line 60-61), but does not adequately explore this later in the results and discussion. If the key ‘story’ is to be centred around this, then it must be discussed more rigorously in the latter sections of the manuscript. The authors should improve this section as a priority, and I am concerned that existing work is omitted from the manuscript. For example, the authors would benefit from reviewing the work of Bunce et al. (2018) where the correlation between the presence of reverse bedrock slopes and retreat rate is clearly established through the analysis of a much larger dataset of marine terminating glaciers (n = 276). Why is this clearly relevant work not cited? How does your work advance understanding in relation to this despite the focus on three sites? Why is fjord geometry not also discussed? Indeed, in your earlier published work you highlight that “Helheim Glacier has retreated and advanced several times throughout the last century; however, likely due to rapid increase of the glacier width and bed elevation farther inland, the retreats have been followed by advances (e.g. during 2005–2006), suggesting both bed and width (see Fig. 11c) play an important role in the Helheim Glacier stability and long-term mass balance” (Khan et al., 2014: 1505). Should your manuscript more clearly establish what is already known about the dynamics of the glaciers studied, to allow you to more clearly communicate the novelty of the work presented here?

Response: It is a clear mistake that Bunce et al. (2018) was not cited. we have added citation to Bunce et al. and Enderlin et al., (2013), which discuss sensitivity of outlet glacier dynamics to shape. Enderlin, E. M., Howat, I. M., and Vieli, A.: High sensitivity of tidewater outlet glacier dynamics to shape, The Cryosphere, 7, 1007–1015, doi:10.5194/tc-7-1007-2013, 2013.

Bunce et al. show higher retreat rates associated with glaciers retreating into widening fjords or reverse bedrock slope.

From line 67 to 72, we discuss the importance of bed and width of the fjord.

We follow the advice from the reviewer and now show contours plot of bed as a separate figure.

Khan et al., 2014, Bunce et al. (2018) and Enderlin et al., (2013) analyse data over the last 2-3 decades. The novelty here is usage of historical data and the expanding the time scale to cover past century and estimation of mass loss since end of the little ice age maximum.

We have added lines in the intro (line 67 to 72) that now cite previous studies to clearly establish what is already known about the dynamics of the glaciers studied.

Moving on, the implications section clearly outlines that mass loss related to climatic forcing is not compensated by stabilising effects associated with changes in sea level and bedrock uplift. The implications of this are briefly reviewed. In the abstract it is argued that “projections may be underestimated and glacier mass loss could exceed worst-case scenario.” The manuscript could be improved by returning to this point in the main body to elaborate in further detail. The details provided do not go significantly beyond the brief summary provided in the abstract.

Response: We have moved the discussion of the stabilizing effects associated with changes in sea level and bedrock uplift from the Method section to the main body of the manuscript. We have also moved old figure 3 from extended data to the main text.

Following the reviewers advise we have extended the discussion about “glacier mass loss could exceed worst-case scenario”. See the new “Discussion” section.

Overall, this manuscript could be published in Nature Communications, however I believe it should first be enhanced with some refinements to the structure, figures, and discussion to sharpen the focus of the paper. I hope the authors find these constructive comments helpful for improving their

manuscript. Below I also append a series of more minor comments on this draft of the manuscript — you may benefit from carefully proof-reading the manuscript to ensure the final draft is ready for publication.

Response: We are very grateful for your insightful response. We have made several changes, which have greatly improved the revised manuscript.

Some specific comments:

Line 26: Responds - change to response?

Response: changed.

Line 30: Suggestion — rephrase the list of glaciers and split this sentence.

Response: Done.

Line 56: Provide a supporting reference.

Response: we have used the basins from Kjeldsen et al (2015) and ice volume from Aschwanden et al 2019. Both are now referred.

Line 60: Provide a reference to support this (RE: the lack of retrograde bed slope at HG). - ref 18, 27?

Response: Ref 18 and 27 added.

Line 127: It is unclear why 'Extended Data Fig. 9' has been cited here.

Response: The purpose of Extended Data Fig. 9 is to show that the results presented in this study agree with others studies. However, we have decided to move the figure to "Method" section.

Line 179: Rephrase this sentence.

Response: Done

Line 155: 'Bump' is rather colloquial.

Response: we have changed the text.

Line 300: Arctic not artic.

Response: Corrected

Line 327: 'i.e.' — Not required?

Response: removed.

Figure 1 — 'Assumed' ice margin? How was this 'assumed'? Please clarify.

Response: This is misleading. We have removed "assumed".

Figure 1 — Please consider adding the source of bed topography here. Are these undulations typically transverse to ice-flow? Note my point regarding this above.

Response: Morlighem M. et al., BedMachine v3: is added as reference and we have added contour plot to better show undulations.

Extended Data Fig. 11 — Caption. Aerial not arial.

Response: corrected.

References cited

Bunce, C., Carr, J.R., Nienow, P.W., Ross, N. and Killick, R., 2018. Ice front change of marine-terminating outlet glaciers in northwest and southeast Greenland during the 21st century. *Journal of Glaciology*, 64(246), 523-535.

Khan, S. A., Kjeldsen, K. K., Kjær, K. H., Bevan, S., Luckman, A., Aschwanden, A., Bjørk, A. A., Korsgaard, N. J., Box, J. E., van den Broeke, M. R., van Dam, T. M., & Fitzner, A., 2014. Glacier dynamics at Helheim and Kangerdlugssuaq glaciers, southeast Greenland, since the Little Ice Age. *Cryosphere*, 8, 1497-1507.

Kjær, K.H., Khan, S.A., Korsgaard, N.J., Wahr, J., Bamber, J.L., Hurkmans, R., van den Broeke, M., Timm, L.H., Kjeldsen, K.K., Bjørk, A.A. and Larsen, N.K., 2012. Aerial photographs reveal late-20th-century dynamic ice loss in northwestern Greenland. *Science*, 337(6094), 569-573.

Kjeldsen, K.K., Korsgaard, N.J., Bjørk, A.A., Khan, S.A., Box, J.E., Funder, S., Larsen, N.K., Bamber, J.L., Colgan, W., Van Den Broeke, M. and Siggaard-Andersen, M.L., 2015. Spatial and temporal distribution of mass loss from the Greenland Ice Sheet since AD 1900. *Nature*, 528(7582), 396-400.